Effects of foliar fertilizer application on the growth and fruit quality of commercial melon varieties grown in a soilless culture system

Khomphet Thanet 1 thanet.kh@mail.wu.ac.th
Promwee Athakorn 1
Islam Shams Shaila 2
1 School of Agricultural Technology and Food Industry, Walailak University , Thasala, Nakhon Si Thammarat , Thailand
2 Department of Agronomy, Hajee Mohammad Danesh Science and Technology University , Basherhat , Bangladesh
Sarkar Sukamal
Electronic publication date: 2023 Feb 21
Publication date: 2023
Volume: 11
Electronic Location ID: e14900
Received 2022 Jul 12; Accepted 2023 Jan 24
Copyright: © 2023 Khomphet et al.
Copyright year: 2023
Copyright holder: Khomphet et al.
License: This is an open access article distributed under the terms of the Creative Commons Attribution License, which permits unrestricted use, distribution, reproduction and adaptation in any medium and for any purpose provided that it is properly attributed. For attribution, the original author(s), title, publication source (PeerJ) and either DOI or URL of the article must be cited.
License URL: https://creativecommons.org/licenses/by/4.0/

Keywords: Melon, Foliar fertilizer, Fruit quality, Fruit growth, Soilless culture system, Melon growth, Melon variety

Funding: Walailak University WU64227, 2021 The work was supported by the personal fund of Walailak University (grant numbers WU64227, 2021). The funders had no role in study design, data collection and analysis, decision to publish, or preparation of the manuscript.

==============================
Several factors influence the quality of melon fruits and foliar fertilizer application is one method for improving their quality. The objectives of this study were: (1) to investigate the response of commercial melon varieties to a soilless culture system in Nakhon Si Thammarat Province, Thailand, and (2) to evaluate the quality of melon fruit under various foliar fertilizer treatments. The experiment was arranged as a completely randomized block design with four replications. Eight commercial melon varieties, including four orange pulp melons (Sandee, Baramee, Sanwan, and Melon cat 697) and four green pulp melons (Kissme, Snowgreen, Melon Princess, and Kimoji), were used in this study. At 1–5 weeks after planting, the growth of the melons was measured using agronomic traits. Four foliar fertilizers (distilled water, micronutrients, secondary nutrients + micronutrients, and amino acid + micronutrients) were sprayed on the melon leaves at 1–5 weeks after pollination, and the growth of the melons, using fruit traits, was recorded. After harvesting, the melons were assessed for the quality of the fruit. This study was conducted at the School of Agricultural Technology and Food Industry’s greenhouse and the Food Chemistry Laboratory of the Center for Scientific and Technological Equipment, Walailak University. In nearly all of the observed growth weeks, the data demonstrate that most agronomic and fruit traits were significantly different between the melon varieties. Sandee, Baramee, Melon cat 697, and Melon Princess are recommended for planting under Nakhon Si Thammarat’s climate, based on fruit size and quality. Foliar fertilizer application impacted the shape, skin color, and quality of the melon. Melons treated with micronutrients, secondary nutrients and micronutrients, and amino acids and micronutrients exhibited better measures of fruit quality than those treated with non-foliar treatments. There was also an interaction observed between melon variety and foliar fertilizer application. Based on measures of fruit quality, Baramee, Melon cat 697, Kissme, and Melon Princess were more responsive to foliar fertilizer application than other melon varieties tested.

Introduction

Melon, or sweet melon (Cucumis melo L.), a member of the Cucurbitaceae family, is one of the world’s most well-known fruit crops and is an excellent source of phytonutrients such as cucurbitacin, lithium, and zinc (Yavuz et al., 2021). Because of their high antioxidant content, anti-inflammatory qualities, antidiabetic advantages, antiulcerogenic activity, and antibacterial capabilities, melons have significant human health benefits (Bianchi et al., 2016). Melon fruit production depends on several factors such as variety, growing season, soil property, pollination, irrigation, and fertilization (Nerson, 2011; Martuscelli et al., 2015). Fertilization is critical to melon fruit quality. Several studies have found that a lack or imbalance of plant nutrients causes melon fruit quality to deteriorate. For example, cracked melons are associated with numerous elements including K, P, Ca, Mg, Mn, Na, B, and Zn (Lopez-Zaplana et al., 2020). Some of the fruit quality attributes of melons, such as fruit firmness, pulp thickness, rind thickness, and TSS, are influenced by nitrogen associated with phosphorus (Bouzo, Céccoli & Muñoz, 2018).

Foliar fertilizer application, where plant minerals are dissolved in a solution and sprayed directly onto plant leaves, is a technique used to improve fruit quality and productivity in a variety of horticulture crops (Muñoz, Ruiz & Bouzo, 2017). Minerals are absorbed through the stomata and epidermis of the leaves (Patil & Chetan, 2018). However, because this procedure may be detrimental to plant leaves, it is only recommended for low-concentration fertilizers. Foliar fertilization and spraying are regularly used in melon cultivation. In one study, the appearance of melon fruits significantly improved following the foliar fertigation of several micronutrients (Lopez-Zaplana et al., 2020). Early flowering, fruit number, and total yield are all promoted by the foliar fertigation of potassium and calcium fertilizers alone, or in combination with borax (Srilatha, Padmodaya & Kumar, 2017). In another study, the foliar spray of some plant growth stimulants, such as plant amino acid and yeast extract, along with nitrogen fertilization, improved some melon plant growth characteristics (plant length, leaf area, weight of shoots per plant) and fruit qualities (fruit diameter, fruit weight, fresh yield, nutrient uptake, total sugar, and TSS) (Morsy, Abdel-Salam & Shams, 2018).

A soilless culture system is a method of growing plants which does not require the use of soil as a growing medium. Plants acquire nutrients by irrigation water, which is referred to as “nutrient solution.” The advantages of a soilless culture system include a reduction or elimination of soil-borne diseases, the ability to control the amount of water and nutrients used, and an increase in nutrient availability (Putra & Yuliando, 2015). This technique is utilized in a wide range of vegetable and horticultural crops, such as lettuce, strawberry, tomato, and melon (Savvas & Gruda, 2018). Melons planted in a soilless culture system have higher values of stem height, leaf number per plant, total leaf number, fruit per plant, fruit weight, and total fruit yield, than melons planted using soil cultivation (Singer et al., 2013). High quality melon fruit production requires appropriate nutrient management. It is crucial for melon cultivation in Southern Thailand to determine the best foliar fertilizer treatment for various melon varieties.

The objectives of this study were: (1) to investigate the response of commercial melon varieties to a soilless culture system in Nakhon Si Thammarat Province, Thailand and (2) to evaluate the quality of melon fruit under various foliar fertilizer applications.

Materials and Methods

Melon varieties and experimental design

This study was conducted in the greenhouse of the School of Agricultural Technology and Food Industry (8°38′43.1″N 99°54′04.4″E) and the Food Chemistry Laboratory of the Center for Scientific and Technological Equipment (8°38′29.5″N 99°53′42.6″E), Walailak University, Nakhon Si Thammarat, Thailand. The greenhouse is 240 square meters, with a width of 8 m and a length of 30 m. Eight marketable melon varieties were chosen for this investigation, including four orange pulp melon varieties: Sandee (V1), Baramee (V2), Sanwan (V3), and Melon cat 697 (V4), and four green melon varieties: Kissme (V5), Snowgreen (V6), Melon Princess (V7), and Kimoji (V8). Melon seeds were sown in a planting tray, using peatmoss as the seed germination medium. Germinated seeds were then transplanted to plastic bags (8 × 13 inch) 10 days after germination, with a 1:1 combination of chopped coconut mesocarp and coconut coir used as the plant medium. The melons were spaced out in a four-block, randomized complete block design (RCBD), with four plants in each block used for data collection. Each melon was given its own automated water application system. Fertigation was carried out using Cosme et al. (2017)’s modified approach. The fertilizer was prepared from calcium nitrate (89.5 g 100 L−1), potassium nitrate (74 g 100 L−1), monoammonium phosphate (19.2 g 100 L−1), magnesium sulfate (40 g 100 L−1), copper sulfate (0.08 g 100 L−1), zinc sulfate (0.14 g 100 L−1), magnesium sulfate (0.19 g 100 L−1), boric acid (0.06 g 100 L−1), sodium molybdate (0.013 g 100 L−1), and Fe-EDTA (1.63 g 100 L−1). Fertilization was applied to the melons four times per day (8 am, 11 am, 2 pm, and 5 pm).

Each of the four groups contained all eight melon varieties, which were each sprayed 1–5 weeks after pollination with 200 ml of one of four possible foliar fertilizer treatments: distilled water (control): C; micronutrients: M (0.134 mM ferrous sulfate (FeSO4)), 0.057 mM zinc sulfate (ZnSO4), 0.096 mM manganese sulfate (MnSO4), 0.090 mM sodium borate (Na2B4O7·5H2O), 0.024 mM copper sulfate (CuSO4), and 0.005 mM sodium molybdate (Na2MoO4); secondary nutrients and micronutrients: S+M (0.214 mM calcium oxide (CaO), 0.275 mM magnesium oxide (MgO), 0.134 mM FeSO4, 0.057 mM ZnSO4, 0.096 mM MnSO4, 0.090 mM Na2B4O7·5H2O, 0.024 mM CuSO4, and 0.005 mM Na2MoO4); or amino acid + micronutrients: A+M (20 gL−1 combined amino acids, 0.134 mM FeSO4, 0.057 mM ZnSO4, 0.096 mM MnSO4, 0.090 mM Na2B4O7·5H2O, 0.024 mM CuSO4, and 0.005 mM Na2MoO4). The combined amino acids were alanine (2.08%), arginine (3.82%), aspartic acid (3.4%), cystine (0.64%), glutamic acid (7.2%), glycine (4.22%), histidine (0.47%), isoleucine (0.99%), leucine (1.05%), lysine (3.15%), methionine (0.31%), phenylalanine (1.14%), proline (3.01%), serine (4.24%), threonine (2.81%), tyrosine (0.47%), and valine (2.01%).

The experiment was conducted between November 2021 and February 2022. During the experiment, the temperature ranged between 21.5 °C and 33.9 °C, with February as the hottest month. Monthly rainfall ranged between 27.6 and 446.8 mm, and the relative humidity ranged between 84 and 93 percent, with the highest amount of rainfall and humidity occurring in November, and the highest relative humidity in February. The weather data was recorded by the greenhouse’s weather station.

Data collection

Plant growth and flowering characteristics, including stem diameter (measured at the middle of stem), stem length, leaf width, leaf length, leaf number, days to male flowering, and days to female flowering, were recorded 1–5 weeks after transplanting. Fruit growth characteristics, including fruit weight, fruit height, and fruit perimeter, were recorded 1–6 weeks after pollination. Ripe melons were harvested at 6 to 7 weeks after pollination for the fruit quality assessment. The fruits were measured and analyzed for fruit cavity size, pulp thickness, rind thickness, firmness, total soluble solid (TSS), juice electrical conductivity (EC), juice pH, titratable acidity (TA), and pulp color (L*, a*, b*, hue, and chroma (c*)). A portable Hunterlab ColorFlex®EZ device (Hunter Associates Laboratory, Reston, VA, USA) was used to collect colorimetric data. A white and black standard were used to calibrate the device. The following measurement modes were chosen: tristimulus L* (lightness), a* (redness/greenness), and b* (yellowness/blueness). The L*-axis measures lightness on a scale of 0 (black) to 100 (white). The a*-axis goes from green (−a*) to red (+a*), while the b*-axis runs from blue (−b*) to yellow (+b*). Hue is defined as the color appearance parameters of a color, also known as color tone or color name. Chroma is a color’s saturation level, with possible chroma values: clear, bright, or brilliant.

A Brix refractometer (model RHB-18ATC; Sinotech Ltd., Watthana, Thailand) was used to measure the TSS, with results expressed as °Brix. TSS measures the balance of sugars and acids in a matrix, which affects fruit flavor (Fundo et al., 2018). EC and pH meters (edge® model HI2030; Hanna Instruments, Inc., Woonsocket, RI, USA) were used to determine the electrical conductivity and pH of the juice. The TA was determined by titrating juice extract to pH 8.1 with 0.1 M NaOH and then measuring the percentage of citric acid in the samples (Piñero et al., 2020).

Data analysis

The variance of melon parameters was calculated using ANOVA. The means were compared using Duncan’s multiple range test (DMRT) and differences were reported as significant at the p < 0.05 or p < 0.01 levels. The melon fruit traits were analyzed using a principal components analysis and the first two components were plotted. All statistical analyses were performed using the R software (version 3.6.1) with the Agricolae and Devtools packages (de Mendiburu, 2021; Wickham et al., 2021).

Results

Melon agronomic traits

Table 1 shows the analysis of variance of agronomic traits among the eight melon varieties. Most agronomic traits were significantly different (p < 0.05 or 0.01) between the melon varieties in all observed weeks, except for stem diameter at the first and third weeks after planting. Figure 1 illustrates melon growth based on agronomic traits. Five weeks after planting, V8 and V7 (1.02 ± 0.02 and 1.00 ± 0.03 cm) produced the biggest stem diameter and V7, V4, V3, and V1 (259.06 ± 7.15, 258.50 ± 7.04, 251.56 ± 6.62, 241.31 ± 5.08 cm) produced the longest stem length. V1 (27.38 ± 0.46 and 24.78 ± 0.41 cm) produced the widest and longest leaf and V2 (38.38 ± 0.52) produced the highest number of leaves. V8 (14.88 ± 0.43 days) had the earliest male flowers and V2, V5, V1, V3, and V7 (23.88 ± 0.29, 24.00 ± 0.43, 24.56 ± 0.27, 24.56 ± 0.42, 24.81 ± 0.61 days) had the earliest female flowers.

Table 1 Analysis of variance of melon agronomic traits.

Sources	Week after planting	Block MS	Treatment MS	Error	C.V. (%)	
Stem diameter (cm)	1st week	0.002ns	0.004ns	0.002	15.70	
2nd week	0.007ns	0.071*	0.006	8.08	
3rd week	0.826ns	0.496ns	0.627	7.16	
4th week	0.011ns	0.022**	0.004	6.23	
5th week	0.045**	0.032**	0.006	8.75	
Stem length (cm)	1st week	4.032ns	12.829**	1.603	20.93	
2nd week	319.800**	932.600**	50.600	21.49	
3rd week	1,649.000**	8,416.000**	181.000	12.52	
4th week	6,716.000**	12,316.000**	155.000	6.98	
5th week	9,728.000**	14,504.000**	401.00	8.77	
Leaf width (cm)	1st week	0.622ns	3.262**	0.765	10.07	
2nd week	21.632**	27.496**	2.424	8.09	
3rd week	40.220**	18.710**	2.030	6.01	
4th week	53.240**	20.870**	1.880	5.49	
5th week	53.360**	26.830**	2.530	6.43	
Leaf length (cm)	1st week	0.976ns	3.220**	0.542	10.50	
2nd week	18.705**	27.285**	2.352	8.57	
3rd week	27.106**	16.687**	1.631	6.15	
4th week	29.995**	11.179**	1.872	6.27	
5th week	29.296**	27.001**	2.207	6.76	
Leaf number (cm)	1st week	0.5208ns	0.9821**	0.200	12.67	
2nd week	0.687ns	8.464**	1.134	9.91	
3rd week	9.438**	30.554**	2.382	8.06	
4th week	25.93**	56.82**	2.420	5.50	
5th week	56.2**	73.23**	3.050	4.91	
Days to male flowering (day)	–	0.448ns	9.853**	1.311	7.18	
Days to female flowering (day)	–	1.271ns	11.750**	2.700	6.01	
Note:

ns is not significantly different, *, ** are significantly different at p < 0.05 and 0.01, respectively.

Figure 1 Melon growth, based on agronomic traits.

(A) Stem diameter, (B) stem length, (C) leaf width, (D) leaf length, (E) leaf number, and (F) days to flowering. V1: Sandee, V2: Baramee, V3: Sanwan, V4: Melon cat 697, V5: Kissme, V6: Snowgreen, V7: Melon Princess, and V8: Kimoji. The black horizontal line in each box indicates the mean. The upper and lower boundaries of the box indicate the 75th and 25th percentiles of the data. The upper and lower whiskers indicate the maximum and minimum data points. The circles above or below the box indicate outlier data points.

Melon fruit traits

Figure 2 illustrates melon fruit characteristics under various foliar fertilizer applications. Melon shape and color intensity are affected by foliar fertilizer application. Table 2 shows the analysis of variance of melon fruit traits. All fruit traits were significantly different (at p < 0.01) between the melon varieties in all observed weeks, except for the melon fruit cavity size. Table 2 shows that most fruit traits were significantly different between foliar fertilizer treatments (at p < 0.05 or 0.01) in most observed weeks except: fruit weight in the first week, fruit height in all observed weeks, fruit perimeter in the first and third weeks, fruit cavity, L* values, and c* values.

Figure 2 Melon fruit characteristics under various foliar fertilizer applications.

V1: Sandee, V2: Baramee, V3: Sanwan, V4: Melon cat 697, V5: Kissme, V6: Snowgreen, V7: Melon Princess, and V8: Kimoji. C: Control, M: Micronutrient, S+M: Secondary nutrient + Micronutrient, and A+M: Amino acid + Micronutrient.

Table 2 Analysis of variance of melon fruit traits.

Sources	Week after pollination	Block MS	Treatment (a) MS	Fertilizer (b) MS	a*b	Error	C.V. (%)	
Fruit weight (kg)	1st week	0.01ns	0.05**	0.01ns	0.01**	0.01	20.83	
2nd week	0.07*	0.42**	0.09**	0.03ns	0.02	13.95	
3rd week	0.25**	0.74**	0.19**	0.04*	0.02	10.82	
4th week	0.57**	1.02**	0.27**	0.07**	0.03	11.31	
5th week	0.68**	1.25**	0.23**	0.07*	0.04	11.48	
6th week	0.41**	0.99**	0.24**	0.14**	0.05	11.40	
Fruit height (cm)	1st week	1.96*	6.32**	0.36ns	1.84**	0.66	9.03	
2nd week	1.88**	9.52**	0.63ns	0.88*	0.44	5.58	
3rd week	7.56**	13.54**	0.03ns	0.70**	0.28	4.11	
4th week	10.25**	16.67**	0.56ns	0.93**	0.33	4.32	
5th week	11.92**	16.93**	0.77ns	1.04**	0.45	4.97	
6th week	5.17**	10.77**	1.70ns	1.45ns	1	4.65	
Fruit perimeter (cm)	1st week	12.00ns	26.35**	1.78ns	12.99**	5.38	9.14	
2nd week	9.79*	61.35**	6.33ns	4.23ns	3.01	4.70	
3rd week	50.36**	127.64**	3.72ns	5.00**	2.12	3.54	
4th week	60.04**	186.46**	9.53*	4.96ns	3	4.08	
5th week	97.81**	197.52**	10.94*	5.67ns	3.49	4.30	
6th week	34.48**	88.18**	17.06*	11.90*	5.25	4.19	
Fruit cavity (cm)	6th week	4.59ns	14.29ns	19.88ns	10.51ns	9.87	60.05	
Pulp thickness (cm)	6th week	0.74**	3.84**	3.01**	0.93**	0.11	9.36	
Rind thickness (cm)	6th week	0.03ns	0.09**	0.03*	0.03**	0.01	23.02	
Firmness (N)	6th week	0.04ns	0.77**	0.17**	0.13**	0.02	16.43	
TSS (°Brix)	6th week	13.58**	24.04**	71.47**	15.17**	1.91	15.15	
EC (µS cm−1)	6th week	2,809,610**	8,031,763**	1,665,111**	1,590,383**	344,379	9.44	
pH	6th week	0.01ns	1.67**	0.28**	0.30**	0.02	2.00	
TA (% citric acid)	6th week	0.01**	0.02**	0.01*	0.01**	0.01	14.44	
L*	6th week	143.60**	729.80**	123.40**	174.80**	25.2	7.64	
a*	6th week	1.00ns	4,384.00**	1.00ns	4.00**	1.00	15.67	
b*	6th week	10.10ns	411.10**	21.50**	9.50**	4.3.	5.97	
hue	6th week	18.00ns	5,688.00**	126.00**	29.00**	14.00	8.02	
c*	6th week	0.03ns	0.10**	0.01ns	0.01ns	0.02	41.27	
Note:

ns is not significantly different, *, ** are significantly different at p < 0.05 and 0.01, respectively.

Figure 3 illustrates melon growth, based on fruit traits. Six weeks after pollination, V7 (2.07 ± 0.09 kg) produced the heaviest fruit while the lowest fruit weight was obtained from V6 (1.00 ± 0.05 kg). V7, V4, V1, V2, and V5 (15.59 ± 0.20, 14.91 ± 0.23, 14.90 ± 0.16, 14.86 ± 0.26, 14.42 ± 0.14 cm) produced the longest fruit and V6 (12.00 ± 0.15 cm) produced the shortest fruit. V7, V5, and V4 (49.25 ± 0.77, 47.50 ± 0.65, 46.35 ± 0.54 cm) produced the fruit with the largest perimeter, and the smallest fruit perimeter was obtained from the fruit of V6 (39.90 ± 0.58 cm). Among foliar fertilizer applications at 6 weeks after pollination, melons treated with A+M (1.64 ± 0.13 kg and 45.69 ± 1.22 cm), S+M (1.59 ± 0.14 kg and 44.94 ± 1.26 cm) and M (1.50 ± 0.09 kg and 44.24 ± 0.79 cm) had larger fruit weights and fruit perimeters than melons under the non-foliar fertilizer application (control: C). There were no significant differences in fruit height between the four foliar fertilizer treatment groups.

Figure 3 Melon growth, based on fruit traits.

(A, D) Fruit weight, (B, E) fruit height, and (C, F) fruit perimeter under various foliar fertilizer applications. V1: Sandee, V2: Baramee, V3: Sanwan, V4: Melon cat 697, V5: Kissme, V6: Snowgreen, V7: Melon Princess, and V8: Kimoji. C: Control, M: Micronutrient, S+M: Secondary nutrient + Micronutrient, and A+M: Amino acid + Micronutrient. The black horizontal line in each box indicates the mean. The upper and lower boundaries of the box indicate the 75th and 25th percentiles of the data. The upper and lower whiskers indicate the maximum and minimum data points. The circles above or below the box indicate outlier data points.

Table 3 shows the mean comparison among the melon fruit traits. Among the melon varieties, V7 produced the thickest (4.16 ± 0.18 cm), firmest (1.11 ± 0.04 N), and brightest (L*; 73.46 ± 0.67) fruit pulp. V4 produced the thinnest fruit rind (0.58 ± 0.03 cm). V3 produced the highest juice EC (6,840.50 ± 113.32 µS cm−1), juice pH (7.08 ± 0.05), and yellowest (b*) pulp (39.93 ± 0.38). V6, V1, V3, V4, and V2 (12.18 ± 0.25, 11.88 ± 0.35, 11.73 ± 0.65, 11.66 ± 0.40, and 11.51 ± 0.38 °Brix) produced the sweetest fruit pulp, and V6 had the highest TA (0.23 ± 0.01% citric acid). Among the foliar fertilizer applications, Table 3 shows that melons treated with foliar fertilizers had better results than control melons for all fruit traits except for fruit cavity size, a*, and c* values, which were not significantly different. Melons treated with M, S+M, and A+M had the thickest (3.75 ± 0.08, 3.63 ± 0.08, 3.69 ± 0.10 cm) and firmest pulp (0.89 ± 0.03, 0.92 ± 0.04, 0.93 ± 0.04 N). Melons treated with S+M had the brightest (L*; 66.02 ± 0.68) pulp. Melons treated with A+M had the sweetest pulp (12.84 ± 0.33), highest juice EC (6,431.46 ± 138.66 µS cm−1), juice pH (6.78 ± 0.03), TA (0.18 ± 0.01), and yellowest (b*) pulp (35.49 ± 0.55).

Table 3 Mean comparison of melon fruit traits.

Treatments	Fruit cavity
(cm)	Pulp thickness
(cm)	Rind thickness
(cm)	Firmness
(N)	TSS
(°Brix)	EC
(µS cm−1)	pH	TA
(% Critic acid)	L*	a*	b*	hue	c*	
V1	4.79 ± 0.17	3.90 ± 0.13b	0.44 ± 0.02cd	0.66 ± 0.04d	11.88 ± 0.35a	6,549.58 ± 175.58ab	6.81 ± 0.04c	0.15 ± 0.01d	62.43 ± 0.70c	18.66 ± 0.40c	36.66 ± 0.48c	31.30 ± 0.56d	0.36 ± 0.03ab	
V2	6.95 ± 0.10	3.51 ± 0.06c	0.49 ± 0.03bc	0.96 ± 0.05b	11.51 ± 0.38a	6,392.08 ± 99.78ab	6.84 ± 0.03bc	0.15 ± 0.01d	63.10 ± 0.59c	19.18 ± 0.41c	38.98 ± 0.39ab	30.66 ± 0.54d	0.36 ± 0.03ab	
V3	4.84 ± 0.12	3.20 ± 0.05d	0.40 ± 0.03d	1.09 ± 0.05a	11.73 ± 0.65a	6,840.50 ± 113.32a	7.08 ± 0.05a	0.18 ± 0.01c	63.11 ± 0.47c	21.55 ± 0.16a	39.93 ± 0.38a	31.55 ± 0.50d	0.40 ± 0.03a	
V4	4.90 ± 0.15	3.76 ± 0.08b	0.58 ± 0.03a	0.81 ± 0.04c	11.66 ± 0.40a	6,170.83 ± 70.31bc	6.92 ± 0.04b	0.16 ± 0.01cd	56.21 ± 3.45d	20.59 ± 0.28b	37.68 ± 0.49bc	32.87 ± 0.80d	0.39 ± 0.03ab	
V5	5.60 ± 0.13	3.79 ± 0.09b	0.49 ± 0.03bc	0.90 ± 0.01bc	10.37 ± 0.30b	5,730.83 ± 109.93c	6.56 ± 0.01d	0.16 ± 0.00d	69.99 ± 0.68ab	−5.37 ± 0.16c	30.41 ± 0.50e	59.61 ± 0.84bc	0.28 ± 0.03bc	
V6	4.50 ± 0.12	3.18 ± 0.09d	0.50 ± 0.02abc	0.93 ± 0.03b	12.18 ± 0.25a	6,601.67 ± 58.43ab	6.64 ± 0.03d	0.23 ± 0.01a	69.21 ± 0.46b	−4.90 ± 0.14c	32.29 ± 0.50d	57.55 ± 0.98c	0.28 ± 0.02bc	
V7	5.35 ± 0.12	4.16 ± 0.18a	0.53 ± 0.03ab	1.11 ± 0.04a	10.14 ± 0.51b	5,039.96 ± 165.98d	6.33 ± 0.08e	0.17 ± 0.00cd	73.46 ± 0.67a	−5.08 ± 0.12c	29.09 ± 0.46e	60.60 ± 1.30ab	0.21 ± 0.02c	
V8	4.93 ± 0.16	3.04 ± 0.08d	0.42 ± 0.02cd	0.62 ± 0.03d	9.40 ± 0.50b	6,418.75 ± 272.73ab	6.38 ± 0.05e	0.21 ± 0.01b	68.48 ± 0.63b	−5.60 ± 0.21c	32.30 ± 0.48d	63.16 ± 0.90a	0.30 ± 0.03abc	
F-test	ns	**	**	**	**	**	**	**	**	**	**	**	**	
C	5.19 ± 0.08	3.20 ± 0.06B	0.51 ± 0.02A	0.80 ± 0.04B	10.13 ± 0.24C	6,316.69 ± 135.29AB	6.60 ± 0.06C	0.18 ± 0.01AB	67.64 ± 0.78AB	7.31 ± 1.83	34.04 ± 0.62B	47.52 ± 2.21A	0.32 ± 0.02	
M	4.75 ± 0.10	3.75 ± 0.08A	0.45 ± 0.02B	0.89 ± 0.03A	10.39 ± 0.28C	6,068.33 ± 113.02B	6.67 ± 0.04BC	0.17 ± 0.01B	63.74 ± 2.01B	7.33 ± 1.84	34.22 ± 0.76B	46.92 ± 2.25A	0.32 ± 0.02	
S+M	4.83 ± 0.11	3.63 ± 0.08A	0.47 ± 0.02AB	0.92 ± 0.04A	11.07 ± 0.32B	6,055.63 ± 124.36B	6.72 ± 0.05AB	0.18 ± 0.01AB	66.02 ± 0.68A	7.29 ± 1.87	34.93 ± 0.64AB	45.25 ± 2.24B	0.33 ± 0.02	
A+M	6.15 ± 0.12	3.69 ± 0.10A	0.49 ± 0.02AB	0.93 ± 0.04A	12.84 ± 0.33A	6,431.46 ± 138.66A	6.78 ± 0.03A	0.18 ± 0.01A	65.59 ± 0.73AB	7.57 ± 1.87	35.49 ± 0.55A	43.96 ± 1.98B	0.31 ± 0.02	
F-test	ns	**	*	**	**	**	**	*	**	ns	**	**	ns	
C: V1	5.12 ± 0.28	3.38 ± 0.10abc	0.42 ± 0.05abc	0.67 ± 0.13bcd	12.57 ± 0.20abc	7,356.67 ± 51.23a	6.63 ± 0.03abc	0.17 ± 0.00ab	62.58 ± 1.08hi	18.99 ± 0.60bcde	37.64 ± 0.78bc	31.87 ± 1.77c	0.43 ± 0.05a	
C: V2	5.37 ± 0.13	3.60 ± 0.15abc	0.67 ± 0.04a	0.76 ± 0.02abcd	10.20 ± 0.00abc	6,450.00 ± 6.32ab	7.02 ± 0.00ab	0.13 ± 0.00ab	61.80 ± 0.22hi	18.42 ± 0.49cde	38.48 ± 0.38abc	32.27 ± 0.79c	0.35 ± 0.07abcd	
C: V3	5.22 ± 0.22	3.10 ± 0.07abc	0.50 ± 0.08abc	0.86 ± 0.04abcd	9.90 ± 0.33abc	6,585.33 ± 209.76ab	7.11 ± 0.03ab	0.14 ± 0.01ab	64.57 ± 0.68efghi	21.36 ± 0.33ab	38.54 ± 0.51abc	31.13 ± 0.60c	0.42 ± 0.04abc	
C: V4	5.12 ± 0.22	3.37 ± 0.13abc	0.63 ± 0.06ab	0.63 ± 0.07bcd	9.82 ± 0.13abc	6,241.67 ± 99.38abc	6.80 ± 0.11abc	0.14 ± 0.00ab	63.90 ± 1.28fghi	20.53 ± 0.47abc	36.30 ± 0.85bcde	36.42 ± 2.56c	0.36 ± 0.06abcd	
C: V5	5.30 ± 0.27	3.68 ± 0.11abc	0.62 ± 0.03abc	0.88 ± 0.02abcd	10.03 ± 0.02abc	5,636.67 ± 177.46abc	6.56 ± 0.03abc	0.16 ± 0.00ab	72.23 ± 0.66abc	−5.44 ± 0.13f	28.66 ± 0.73hi	61.72 ± 1.18ab	0.21 ± 0.02bcd	
C: V6	4.95 ± 0.16	3.00 ± 0.06abc	0.48 ± 0.03abc	0.84 ± 0.09abcd	10.55 ± 0.23abc	6,240.00 ± 76.11abc	6.62 ± 0.03abc	0.22 ± 0.01ab	70.59 ± 0.55bcd	−5.09 ± 0.12f	31.06 ± 0.53ghi	57.84 ± 1.45ab	0.31 ± 0.03abcd	
C: V7	5.45 ± 0.20	2.92 ± 0.20bc	0.40 ± 0.04bc	1.22 ± 0.11abc	7.93 ± 0.39bc	5,061.50 ± 135.47abc	5.84 ± 0.04c	0.18 ± 0.01ab	75.95 ± 1.51a	−5.07 ± 0.25f	29.57 ± 0.37ghi	64.98 ± 1.61a	0.20 ± 0.03cd	
C: V8	5.03 ± 0.28	2.55 ± 0.08c	0.40 ± 0.04bc	0.54 ± 0.03d	10.05 ± 1.28abc	6,961.67 ± 732.76ab	6.24 ± 0.11bc	0.26 ± 0.03a	69.53 ± 1.45bcd	−5.24 ± 0.63f	32.06 ± 1.51fghi	63.92 ± 1.21a	0.31 ± 0.06abcd	
M: V1	4.62 ± 0.36	4.78 ± 0.17a	0.42 ± 0.05abc	0.71 ± 0.10bcd	9.60 ± 0.36abc	5,955.00 ± 217.02abc	6.74 ± 0.05abc	0.13 ± 0.02ab	64.73 ± 1.20efghi	17.03 ± 0.57e	35.64 ± 1.41cdef	31.01 ± 0.45c	0.24 ± 0.03abcd	
M: V2	4.90 ± 0.12	3.45 ± 0.15abc	0.42 ± 0.03abc	0.72 ± 0.03abcd	9.72 ± 0.16abc	6,083.33 ± 151.96abc	6.87 ± 0.01ab	0.13 ± 0.00ab	65.90 ± 0.57defgh	20.13 ± 0.67abcd	39.31 ± 0.53abc	32.19 ± 1.12c	0.40 ± 0.07abc	
M: V3	4.65 ± 0.13	3.30 ± 0.11abc	0.35 ± 0.03c	1.12 ± 0.07abcd	13.65 ± 0.25a	7,175.00 ± 251.19ab	6.83 ± 0.13abc	0.21 ± 0.01ab	63.77 ± 0.76fghi	22.02 ± 0.31a	41.96 ± 0.54a	33.48 ± 1.54c	0.37 ± 0.07abcd	
M: V4	4.00 ± 0.19	3.80 ± 0.13abc	0.57 ± 0.06abc	0.86 ± 0.01abcd	10.03 ± 0.02abc	5,676.67 ± 2.11abc	7.06 ± 0.00ab	0.12 ± 0.00b	64.00 ± 9.60efghi	20.25 ± 0.37abcd	36.98 ± 0.49bcd	31.68 ± 0.54c	0.39 ± 0.08abcd	
M: V5	5.62 ± 0.21	3.57 ± 0.09abc	0.38 ± 0.05bc	0.91 ± 0.02abcd	11.20 ± 0.16abc	6,416.67 ± 8.43ab	6.57 ± 0.04abc	0.18 ± 0.00ab	68.96 ± 1.33bcde	−4.42 ± 0.23f	28.81 ± 0.83hi	60.84 ± 2.15ab	0.31 ± 0.05abcd	
M: V6	4.07 ± 0.17	3.35 ± 0.08abc	0.57 ± 0.02abc	0.97 ± 0.01abcd	12.15 ± 0.49abc	6,925.00 ± 42.64ab	6.51 ± 0.00abc	0.23 ± 0.01ab	69.75 ± 1.01bcd	−5.31 ± 0.28f	32.37 ± 1.50efgh	58.62 ± 3.10ab	0.26 ± 0.05abcd	
M: V7	5.45 ± 0.15	4.30 ± 0.14abc	0.53 ± 0.03abc	1.11 ± 0.06abcd	9.53 ± 0.22abc	5,005.00 ± 161.76bc	6.28 ± 0.04bc	0.18 ± 0.00ab	73.52 ± 1.40ab	−5.13 ± 0.16f	28.01 ± 0.80i	62.36 ± 1.59a	0.24 ± 0.05abcd	
M: V8	4.82 ± 0.11	3.45 ± 0.08abc	0.40 ± 0.04bc	0.70 ± 0.04bcd	7.25 ± 0.11c	5,310.00 ± 93.99abc	6.54 ± 0.00abc	0.15 ± 0.01ab	68.46 ± 0.88cdef	−5.90 ± 0.15f	30.65 ± 0.82ghi	65.17 ± 1.69a	0.36 ± 0.06abcd	
S+M: V1	4.02 ± 0.24	3.83 ± 0.16abc	0.43 ± 0.05abc	0.59 ± 0.03cd	11.57 ± 0.17abc	5,886.67 ± 61.19abc	6.91 ± 0.03ab	0.14 ± 0.01ab	62.08 ± 1.22hi	18.93 ± 0.94bcde	36.29 ± 0.93bcde	29.47 ± 0.95c	0.38 ± 0.06abcd	
S+M: V2	5.13 ± 0.27	3.57 ± 0.14abc	0.40 ± 0.04bc	1.28 ± 0.03ab	12.00 ± 0.11abc	6,560.00 ± 38.82ab	6.58 ± 0.01abc	0.18 ± 0.00ab	62.17 ± 1.18hi	17.78 ± 0.99de	37.69 ± 1.21bc	29.04 ± 0.88c	0.33 ± 0.08abcd	
S+M: V3	4.60 ± 0.32	3.22 ± 0.09abc	0.37 ± 0.05bc	1.01 ± 0.03abcd	8.95 ± 1.75abc	6,418.33 ± 175.18ab	7.32 ± 0.04a	0.15 ± 0.00ab	62.64 ± 0.50hi	21.29 ± 0.24ab	39.81 ± 0.59abc	31.41 ± 0.58c	0.45 ± 0.03a	
S+M: V4	5.17 ± 0.20	4.02 ± 0.11abc	0.57 ± 0.06abc	0.80 ± 0.11abcd	13.15 ± 0.25ab	6,393.33 ± 43.41ab	7.06 ± 0.00ab	0.20 ± 0.01ab	62.63 ± 1.30hi	21.89 ± 0.34a	39.88 ± 0.54abc	31.33 ± 1.02c	0.43 ± 0.08ab	
S+M: V5	5.37 ± 0.21	3.52 ± 0.10abc	0.49 ± 0.06abc	0.86 ± 0.04abcd	10.67 ± 0.89abc	5,695.00 ± 8.85abc	6.55 ± 0.02abc	0.14 ± 0.00ab	69.10 ± 1.26bcde	−5.97 ± 0.33f	32.24 ± 0.65efgh	58.45 ± 1.06ab	0.36 ± 0.06abcd	
S+M: V6	4.70 ± 0.17	3.32 ± 0.15abc	0.48 ± 0.06abc	0.97 ± 0.02abcd	12.98 ± 0.10ab	6,671.67 ± 81.38ab	6.81 ± 0.03abc	0.24 ± 0.01ab	68.17 ± 0.40cdef	−4.70 ± 0.37f	32.20 ± 1.01efgh	56.27 ± 1.66ab	0.21 ± 0.05bcd	
S+M: V7	5.23 ± 0.32	4.65 ± 0.15ab	0.63 ± 0.05ab	1.12 ± 0.07abcd	9.15 ± 0.49abc	4,125.00 ± 288.48c	6.31 ± 0.09abc	0.14 ± 0.01ab	73.27 ± 0.83ab	−5.28 ± 0.29f	28.01 ± 1.34i	62.47 ± 1.83a	0.22 ± 0.03abcd	
S+M: V8	4.40 ± 0.27	2.90 ± 0.10bc	0.42 ± 0.03abc	0.72 ± 0.04abcd	10.12 ± 0.08abc	6,695.00 ± 17.84ab	6.27 ± 0.02bc	0.22 ± 0.01ab	68.11 ± 1.68cdef	−5.59 ± 0.20f	33.34 ± 0.50defg	63.59 ± 0.61a	0.31 ± 0.07abcd	
A+M: V1	5.40 ± 0.26	3.62 ± 0.11abc	0.50 ± 0.04abc	0.69 ± 0.07bcd	13.77 ± 0.40a	7,000.00 ± 429.34ab	6.96 ± 0.09ab	0.18 ± 0.01ab	60.35 ± 1.76i	19.67 ± 0.74abcd	37.07 ± 0.63bcd	32.84 ± 0.61c	0.38 ± 0.04abcd	
A+M: V2	5.18 ± 0.22	3.42 ± 0.07abc	0.47 ± 0.04abc	1.09 ± 0.08abcd	14.13 ± 0.44a	6,475.00 ± 363.07ab	6.89 ± 0.02ab	0.17 ± 0.01ab	62.53 ± 1.54hi	20.38 ± 0.62abcd	40.45 ± 0.30ab	29.14 ± 0.84c	0.36 ± 0.05abcd	
A+M: V3	4.88 ± 0.25	3.18 ± 0.09abc	0.37 ± 0.03bc	1.37 ± 0.07a	14.42 ± 0.34a	7,183.33 ± 62.65ab	7.07 ± 0.04ab	0.22 ± 0.01ab	61.47 ± 1.31hi	21.53 ± 0.41ab	39.44 ± 0.69abc	30.17 ± 0.66c	0.36 ± 0.05abcd	
A+M: V4	5.40 ± 0.15	3.87 ± 0.15abc	0.57 ± 0.04abc	0.95 ± 0.01abcd	13.65 ± 0.66a	6,371.67 ± 106.66abc	6.77 ± 0.06abc	0.19 ± 0.01ab	63.42 ± 1.40ghi	19.69 ± 0.70abcd	37.57 ± 1.33bc	32.06 ± 0.75c	0.37 ± 0.06abcd	
A+M: V5	5.90 ± 0.28	4.40 ± 0.14ab	0.47 ± 0.04abc	0.94 ± 0.01abcd	9.58 ± 0.73abc	5,175.00 ± 181.16abc	6.58 ± 0.02abc	0.14 ± 0.01ab	69.67 ± 1.80bcd	−5.66 ± 0.22f	31.95 ± 0.88fghi	57.44 ± 1.84ab	0.23 ± 0.05abcd	
A+M: V6	4.30 ± 0.31	3.03 ± 0.30abc	0.48 ± 0.06abc	0.92 ± 0.03abcd	13.05 ± 0.18ab	6,570.00 ± 18.26ab	6.63 ± 0.02abc	0.23 ± 0.01ab	68.34 ± 1.24cdef	−4.51 ± 0.27f	33.51 ± 0.69defg	57.48 ± 1.62ab	0.33 ± 0.04abcd	
A+M: V7	5.27 ± 0.28	4.77 ± 0.23a	0.57 ± 0.04abc	1.00 ± 0.05abcd	13.95 ± 0.44a	5,968.33 ± 197.58abc	6.90 ± 0.02ab	0.16 ± 0.01ab	71.10 ± 0.97bc	−4.86 ± 0.28f	30.78 ± 0.44ghi	52.57 ± 2.16b	0.17 ± 0.03b	
A+M: V8	5.48 ± 0.45	3.27 ± 0.11abc	0.47 ± 0.02abc	0.53 ± 0.05d	10.17 ± 1.31abc	6,708.33 ± 696.91ab	6.47 ± 0.11abc	0.19 ± 0.02ab	67.81 ± 1.07cdefg	−5.67 ± 0.56f	33.15 ± 0.43defg	59.96 ± 2.66ab	0.25 ± 0.05abcd	
F-test	ns	**	**	**	**	**	**	**	**	**	**	**	**	
Note:

ns is not significantly different, *, ** are significantly different at p < 0.05 and 0.01, respectively. Values with the same alphabetical superscript within the same column are not significantly different based on Duncan’s multiple range test.

Figure 4 illustrates the biplot between the first two principal components of melon fruit quality under various foliar fertilizer applications. V7 is the largest cluster in the first principal component and dominates in L*, hue, and fruit cavity size, while V6 is the smallest cluster and dominates in TA. The fruits of V1–V4 dominate EC, TSS, a*, b*, and c*.

Figure 4 Biplot between the first two PCs of melon fruit quality under various foliar fertilizer applications.

V1: Sandee, V2: Baramee, V3: Sanwan, V4: Melon cat 697, V5: Kissme, V6: Snowgreen, V7: Melon Princess, and V8: Kimoji. C: Control, M: Micronutrient, S+M: Secondary nutrient + Micronutrient, and A+M: Amino acid + Micronutrient. PC1 and PC2 are the first and second principal components. The clusters represent the distribution of melon fruit quality under various foliar fertilizer applications. Arrows indicate the direction of melon fruit quality characteristics.

Discussion

This study shows that most melon agronomic and fruit traits are significantly different between melon varieties in all observed growth periods. Melons treated with foliar fertilizers had better measures of fruit quality including: fruit weight, fruit height, fruit perimeter, pulp thickness, rind thickness, firmness, TSS, EC, TA, and pulp colors. According to Zaniewicz-Bajkowska et al. (2010), foliar feeding is an effective method of supplying nutrients during the period of intensive plant growth when it can most improve the mineral status of the plants and increase crop yield. Melon is a polymorphic taxon with many botanical and physiological varieties. Melon fruits also have a wide range of features, such as color, shape, size, skin pattern, sweetness level, and odor (Lima & Beevy, 2021).

Foliar fertilizer application is widely used as a technique to improve the fruit quality and productivity of horticultural crops (Santos, 2013). This type of direct application can help to reduce the overall quantity of fertilizer needed in plant production while preserving fertilizer efficiency. Furthermore, foliar fertilizer application helps minimize the adverse consequences of excessive fertilizer usage, such as soil acidification, salinization, and nutrient unavailability (Niu et al., 2021). Foliar fertilizer also impacts the quality of melon fruits. To prevent melon fruit cracking, which has been linked to a number of elements (B, Ca, K, Mg, Mn, Na, P, and Zn), Lopez-Zaplana et al. (2020) utilized calcium fertilizers, micronutrient fertilizers, and a mixture of calcium and micronutrient fertilizers. Another study, Muñoz, Ruiz & Bouzo (2017) sprayed varying quantities of calcium nitrate every week after fruit set and found that fruits were firmer at 2.6 and 5.2 g L−1 calcium nitrate concentrations. Srilatha, Padmodaya & Kumar (2017) applied a mixture of borax, potassium nitrate, and calcium nitrate and found the combination was effective in promoting early flowering, fruit number, and fruit yield.

The resistance and hardness of melon fruit skin are both influenced by calcium, which, as a molecular signaling agent, is a mineral that helps to strengthen plant cell walls and fruit skin (Cybulska, Zdunek & Konstankiewicz, 2011). Magnesium (Mg) is an important component of the chlorophyll molecule, which is necessary for photosynthesis and protein synthesis in plants. Mg has been associated with other elements in melon fruit, such as calcium, potassium, and manganese, according to several studies. Mg is also a cofactor for several enzymes involved in cell wall formation, including glutamine synthetase, xylose isomerase, and isocitrate lyase (Lopez-Zaplana et al., 2020). Iron is a necessary micronutrient for plant metabolism, including photosynthesis, respiration, DNA synthesis, and metabolic activators in a variety of pathways. It is a component of a number of key enzymes, including cytochromes, which are a crucial component of the electron transport chain in cellular respiration (Rout & Sahoo, 2015). Zinc is engaged in the ion transport mechanism in plant cells. Zinc maintains the balance between the phospholipid levels and membrane integrity, which affects plant water absorption and racking (Dang et al., 2010). Manganese also has an impact on various plant cell functions, such as amino acid synthesis and lignin biosynthesis. The hardness of the melon fruit peel is affected by low manganese levels, resulting in fruit cracking (Chen et al., 2016). Boron is required for the formation of new tissues and the production of cell walls. It has a relationship with the cell membrane’s integrity and permeability. Higher calcium levels are produced by an increase in Boron in the melon pulp and rind (Lewis, 2019; Lopez-Zaplana et al., 2020). Copper is an essential element of cellular physiological activities, such as energy generation. It is often used as a foliar fertilizer and stays on the leaf surface. Through antagonist and synergistic actions, excess copper has a negative impact on root metabolic activity and nutrient absorption (Torre, Iovino & Caradonia, 2018). Molybdenum is a trace element that plants need for development and for a variety of metabolic processes. It regulates the oxidation and reduction reactions of enzymes, and is an essential component of organic molecules known as molybdenum co-factors (Kaiser et al., 2005). Plant amino acids are precursors of a number of chemicals that are involved in cell activity. They are involved in the production of nitrogenous bases (purine and pyrimidine), alkaloids, and terpenoids. These compounds are required for pollination and fruit formation (Morsy, Abdel-Salam & Shams, 2018). In future investigations, other fruit quality attributes, such as vitamins and non-vitamin phytochemicals like flavonoids, should be included as they might also be affected by foliar fertilizer application.

Conclusions

Most agronomic traits were significantly different between the melon varieties in all observed growth weeks, except for stem diameter in the first and the third weeks after planting. All fruit traits were significantly different between the melon varieties in all observed growth periods. Sandee, Baramee, Melon cat 697, and Melon Princess are recommended for planting under Nakhon Si Thammarat’s climate, based on fruit size and quality. Most fruit traits were significantly different between the different foliar fertilizer treatment groups in most observed growth weeks. The shape, skin color, and quality of the melon were all affected by foliar fertilizer application. Melons treated with micronutrients, secondary nutrients and micronutrients, and amino acids and micronutrients had higher measures of fruit quality than melons grown with non-foliar applications. There were also interactions observed between melon variety and foliar fertilizer application. Based on measures of fruit quality, Baramee, Melon cat 697, Kissme, and Melon Princess were more responsive to foliar fertilizer application than other melon varieties tested.

Supplemental Information

Supplemental Information 1 Raw data.

Plant growth, fruit growth, and fruit analysis of melons.

Click here for additional data file.

We would like to thank the School of Agriculture Technology and Food Industry for the use of their greenhouse, and the Food Chemistry Laboratory of the Center for Scientific and Technological Equipment, Walailak University for the use of their laboratory equipment.

Additional Information and Declarations

Competing Interests

Author Contributions

Data Availability

The authors declare that they have no competing interests.

Thanet Khomphet conceived and designed the experiments, performed the experiments, analyzed the data, prepared figures and/or tables, authored or reviewed drafts of the article, and approved the final draft.

Athakorn Promwee performed the experiments, authored or reviewed drafts of the article, and approved the final draft.

Shams Shaila Islam analyzed the data, authored or reviewed drafts of the article, and approved the final draft.

The following information was supplied regarding data availability:

The raw data is available in the Supplemental File.

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
