# Peer review of "Effects of foliar fertilizer application on the growth and fruit quality of commercial melon varieties grown in a soilless culture system"

_PeerJ, doi:10.7717/peerj.14900_

## Round 0.1 · original submission · Major Revisions

The manuscript has been extensively reviewed by the four external reviewers and they have provided some critical comments and suggestions for improvement of this manuscript. All have them have found that article has scientific merit but need to improve the article significantly. One of the reviewers has raised question about the presentation and structural setup and other one statistical and data analysis. The article need to improve significantly and thus required a major revision. Please provide point wise explanation for each reviewer while revision of this manuscript.

·

Basic reporting

In. this study, authors show that the use of foliar fertilizer improves agronomic and fruit traits in melon.

Though this is useful study in agriculture sciences, I have several concerns, which I have pointed out below.

None of the figures and tables have proper legend good enough to understand them, or the details on the plot such as axis labels, what bar represents etc. So, I did not understand them in depth. I recommend authors to please add enough description to figures to understand what is depicted and what it means in biological sense. Please replace V1….Vn labels with meaningful words, I also do not understand what are labels on the top of bars. I suggest using boxplots or some alternative graphs to visualize the data since, there are many variables used, and bar plots does not suite for such analysis. Furthermore, why some measurements increase over weeks, and then drops at the last week.


Authors found that the agronomic and fruit traits differ significantly among melon varieties, I wonder whether foliar fertilizer, and nutrient solution analyses account for those differences. Authors need to describe thoroughly in result section.

Lines 216-244 lines in discussion reiterate same prospects for all micronutrients and appears highly speculative. I recommend rephrase this section entirely with the findings obvious from the results obtained in this study than the speculative text about each micronutreint for which authors do not have evidence pertinent to this study.

The flow of introduction and the last paragraph appears totally incoherent. Authors need to bridge the gap.

Several grammatical mistakes to point out but I suggest authors to revise manuscript entirely and proofread for language. Some of the example have been pointed out below and suggested some modifications.

Line 49 – “regarded provide” does not make sense
Lines 53-54 need rephrasing
Line 56 “ influenced by nitrogen associated with phosphorus”, what do you mean ?

Line 130 Its difficult to understand what is “(L*, a*, b*, hue, and c*)”, and even in result section, and I recommend replacing them with meaningful names.

Line 140-141 “The melon fruit traits were analyzed in principal components and the first two components were plotted. All statistical analyses were calculated using R”, please replace with “The melon fruit traits were analyzed using principal components. All statistical analyses were performed using R”

Line 197 replace “significantly different” with “significant differences”

Experimental design

Experimental design is fine. But the representation of results could be improved significantly. I suggests authors to use alternative data visualisation methods since there are lot of variables to use bar plots. Boxplots could be good alternative.

Validity of the findings

In my opinion discussion and conclusions about findings reported in this study is highly speculative and need entire revision as stated above. I suggests to use permutation and bootstrap resampling techniques to validate findings against chance.

Reviewer 2 ·

Basic reporting

Review peerJ-75391
The authors reported the foliar fertilizer on growth and fruit quality of commercial melon varieties under soilless culture system. The work itself is meaningful for soilless fruit growth, not just melon itself. The discussion needs further significant improvement. First, authors should provide more science behind these changes, not just described the results directly; second, the fruit dimensions and related physicochemical properties need to be discussed integral. They should be related to each other via metabolites via metabolism/catabolism, which should be discussed from this viewpoint, for instance, Critical Reviews in Food Science and Nutrition, 61, 1448-1469.
The relationship between foliar fertilizer and the final quality of fruit should be discussed further. For instance, Acta Scientiarum Plonorum-Hortorum Cultus, 9, 55-63.
The quality attributes should be defined clearly. Some physicochemical properties might not be within the category of ‘fruit quality’. For instance, Food Chemistry, 394, 133533; Food Control, 101, 241-250.
For fruit quality, vitamins and non-vitamin phytochemicals like flavonoids are important, which should be discussed further. Under this fertilizer, whether it might be better to enhance the generating of flavonoids? For instance, Food Chemistry, 399, 133959.

Experimental design

The experimental design was good.

Validity of the findings

The findings was valid from the results and explanations.

Additional comments

More discussion should be performed to integrate the results from different parameters.

Reviewer 3 ·

Basic reporting

a. Abstract should be continuous, no need of splitting into background, methods and results
b. Abstract must contain data regarding changes of important parameters.
c. Line 46- correct the sentence as…….an excellent source of phytonutrients such as cucurbitacin, lithium, and zinc………
d. In the para ending at Line no. 78, mention total as well as soil less melon cultivation status in your area and world statistic also.
e. Delete the sentence at Line 82-83: “The research findings may be beneficial to farmers for enhancing melon fruit quality and researchers for further research”.
f. Incase of secondary and micro nutrient solution preparation, did you follow any standard protocol in choosing the concentration of the elements, then cite it or if it is your own determination then specify the reason.
g. Mention the green house type and GPS location in which the experiment conducted
h. Elaborate the working principle and operational details of the semi-automated water application system used to fertigate the melon plants in the experiment. It would be more prominent if you provide a picture of the fertigation system used in your soil less melon cultivation.
i. Is the combine amino acids used in the experiment, prepared by you in laboratory or a commercial product? Mention it
j. In line 131: Brix reflectometer should be replaced with Brix refractometer and if possible, mention the model number of the same.
k. Writing style in result need exhaustive correction…you simply mentioned the data for specific trait but those things anybody found in the table or figure, so, you have to write the thing in a precise manner i.e., which treatments/variety/varieties showed maximum value and % difference from the minimum value recorder.
l. In discussion portion you just mentioned the role of individual micro-nutrient what does it sense? Re write discussion section, it needs only logical information or supporting evidence for each trait you mentioned in the table or figure
m. You, did not raise any specific conclusion. Modify the conclusion with very precise conclusion-i.e., which variety is best in respect to overall parameters? which foliar treatment is best for melon variety and finally the best interaction effect

Experimental design

No comment

Validity of the findings

No comment

Additional comments

No comment

Reviewer 4 ·

Basic reporting

In summary, this manuscript collected abundant data on applying different foliar fertilizer combinations on different varieties of melons. However, extensive statistical analyses were needed. More meaningful and useful results can be included. For example, authors stated in conclusion that many morphology and quality parameters of melons were significantly affected by foliar fertilizer, but authors didn’t mention whether positive or negative effects were caused by foliar fertilizer. There was difference among melon varieties, again, authors didn’t mention which variety had higher qualities compared to others in conclusions. Authors applied different combinations of foliar fertilizers, which one is better?
English language in this manuscript needs extensive improvement.

Experimental design

Please find in additional comments.

Validity of the findings

Please find in additional comments.

Additional comments

General comments
1. Four blocks were included. How many plants were included in each experimental unit? For the data collection, for example, stem diameter data was collected from how many plants within each experimental unit? Same questions for all parameters measured.
2. I recommend authors use cultivar names instead of V1-V8 in tables, figures, results, and discussion. It is difficult to come back and match which variety it is. Another reason is that the variety names are not very long.
3. Figure 3: What is the meaning of asterisk symbol? Please also explain the meaning of different letters on the top of bars (different letters represent significant difference between treatment or time) in legends.
4. In Table 1 and 2, many significant differences can be found in blocks. Why there were so many significant differences among different blocks? Please include discussion on this topic.
5. For statistical analyses, mean separation on each treatment combination is needed in the results. For example, there is significant difference existed among foliar fertilizer, through mean separation, we want to know foliar fertilizer had significant effects on which melon variety/varieties. Mean separation can be done at least on parameters collected in the 5th or 6th week. So, growers will know on which varieties, applying foliar fertilizer might gain better profits, whereas on other varieties might not.

Specific comments
Line 39: ‘secondary and micronutrients’, is there a typo? Why do you mean by ‘secondary nutrients?=7 543 Line 63: For what reasons?
Line 94: Did authors mean some plants grew from seeds; some grew from sprouts?
Line 107-111: First, please use the full name of the element before you use the abbreviations. Second, what salt was used in the foliar fertilizer solution? Metal itself won’t be soluble in water. Third, why ‘M’s in line 107, 108 and 110 have different contents? Fourth, why did you use these concentrations of these elements in foliar fertilizers?
Line 153: widest instead of wildest
Line 163: ‘Table 2 shows’, or ‘from Table 2, it was shown’ instead of ‘the table shows’, similar in line 183.
Line 169: longest instead of tallest
Line 171: How did you measure the perimeter of a melon? Did you measure the longest perimeter around a melon? Please explain detailed measure assay in materials and methods. The same recommendation on other parameters.
Lines 175-176: Please improve English language of this sentence.
Line 253: grave? English in this part need

---

## Round 0.2 · Major Revisions

I am not convinced with the revision made by the authors. One of the reviewers also pointed out several shortfalls of the article, that must have to be addressed. Authors need a substantial revision of the manuscript based on the review comments and also give point wise explanations with possible logical reasons. The English language also needs to improve.

Reviewer 4 ·

Basic reporting

Authors did not revise this manuscript carefully as per all reviewers' comments. English language still needs extensive improvement.

Experimental design

no comment

Validity of the findings

no comment

Additional comments

‘There was difference among melon varieties, again, authors didn’t mention which variety had higher qualities compared to others in conclusions. Authors applied different combinations of foliar fertilizers, which one is better?
Authors: We added the best variety and the interaction between varieties and foliar fertilizer applications’
I didn’t find the best variety and the interaction between interactions. Please add to the abstract and please address which line was amended when you answer each question.

‘5. For statistical analyses, mean separation on each treatment combination is needed in the results. For example, there is significant difference existed among foliar fertilizer, through mean separation, we want to know foliar fertilizer had significant effects on which melon variety/varieties. Mean separation can be done at least on parameters collected in the 5th or 6th week. So, growers will know on which varieties, applying foliar fertilizer might gain better profits, whereas on other varieties might not.
Authors: Because we have so many data and it might make reader confused if we separate each treatment combination, to make it easier, we show the combination as principal component analysis (figure 4). From this figure, reader can see the response of each variety on different foliar fertilizer applications.’
‘Line 107-111: First, please use the full name of the element before you use the abbreviations. Second, what salt was used in the foliar fertilizer solution? Metal itself won’t be soluble in water. Third, why ‘M’s in line 107, 108 and 110 have different contents? Fourth, why did you use these concentrations of these elements in foliar fertilizers?
Authors: The solution used for foliar fertilizer application was prepared at low concentrations. There was no problem with dissolution of element preparation. The concentrations in this study were the common recommendations from our laboratory, however, there was no scientific report on the reason why use that concentration.’
‘Line 171: How did you measure the perimeter of a melon? Did you measure the longest perimeter around a melon? Please explain detailed measure assay in materials and methods. The same recommendation on other parameters.
Authors: The word “longest perimeter” was repeated by “biggest perimeter”. For detailed measurement assay, in our opinion, there is no need to add the detail for measurement the melon parameters because it is a same measurement use in general research on melons or related crops.’

However, authors didn’t respond my questions directly, or gave enough evidence convincing me that they are right. I can not accept their responses to these questions/comments.
Besides, English language in this manuscript still needs extensive improvements.

---

## Round 0.3 · accepted · Accept

The authors have revised the manuscript following all necessary comments of the reviewers. The quality of the manuscript is improved considerably. The manuscript is now acceptable for publication.

Reviewer 4 ·

Basic reporting

Authors had thoroughly responded to all the comments I addressed. I appreciate their efforts on improving the quality of this manuscript.

Experimental design

no comment

Validity of the findings

no comment